# Feasibility of the Xemio app for breast cancer survivors in a clinical setting: Adherence, acceptance, and side effect monitoring (CTCAE vs. QoL)

Maria-Angeles Fuentes-Expósito[1,2]ᵒ, Santiago Frid[3]ᵒ*, Montserrat Muñoz-Mateu[4]ᵒ, Antoni Sisó-Almirall[5]ᵒ, Manuel Armayones Ruiz[2]ᵒ, Inmaculada Grau-Corral[1,6]ᵒ

1 Fundación iSYS, Barcelona, Spain, 2 Universitat Oberta de Catalunya, Barcelona, Spain, 3 Clinical Informatics Service, Hospital Clínic de Barcelona, Barcelona, Spain, 4 Oncology Service, Hospital Clínic de Barcelona, Barcelona, Spain, 5 Department of Medicine, Universitat de Barcelona, Barcelona, Spain 6 mHealth and Digital Health Observatory, Hospital Clínic de Barcelona, Barcelona, Spain

ᵒ These authors contributed equally to this work.
* FRID@clinic.cat

## Abstract

Breast cancer is the most common cancer worldwide, posing significant challenges for survivors, including long-term physical, emotional, and cognitive effects. Mobile health (mHealth) tools provide new opportunities to support these patients by enabling symptom tracking, side effect management, and personalized interventions. This study evaluated the feasibility and acceptability of the Xemio-Research mHealth application as a digital support tool for breast cancer survivors in a clinical setting. It assessed user adherence, system usability, and patient experience. A secondary objective was to compare self-reported side effects in the app with traditional quality-of-life questionnaires. This prospective study was conducted over one year within a European research project. Breast cancer survivors were recruited from a clinical setting, where they installed the Xemio-Research app and were guided on its use. During the study period, participants tracked symptoms, reported side effects, and engaged with the app. Adherence was measured through interaction logs and activity tracking. Usability was assessed using a validated scale, and patient feedback was collected through structured and open-ended survey questions. Among 61 enrolled participants, 49 actively used the app. Adherence was high in the first three months (96%) but declined to 35% by the final trimester. Usability was rated as excellent (82.78/100), and 87% of respondents recommended the app. The app enabled more detailed symptom tracking compared to traditional quality-of-life questionnaires, particularly for joint pain, tingling, and muscle weakness. The Xemio-Research app demonstrated feasibility and acceptability for breast cancer survivors, offering valuable insights into patient-reported outcomes and side effect management. However, sustaining long-term engagement remains a challenge. Integrating real-time

**Data availability statement:** Individual-level data cannot be shared publicly because they contain potentially identifying clinical and sociodemographic information and are protected under the EU General Data Protection Regulation (GDPR), the Spanish Organic Law 3/2018, and the study protocol approved by the Ethics Committee for Drug Research of Hospital Clínic de Barcelona (CEIm-HCB; protocol HCB/2020/0971). Qualified researchers may request access to a minimally necessary, de-identified dataset by contacting CEIm-HCB at CEIm@clinic.cat. Requests will be evaluated for compliance with GDPR, the approved protocol and consent, and patient confidentiality safeguards; if approved, data will be shared under a Data Use Agreement.

**Funding:** This study was supported by Hospital Clínic de Barcelona in the form of a grant awarded to I.G-C, M.M-M and S.F through the EU-funded Research and Innovation Action Artificial Intelligence Supporting Cancer Patients across Europe (ASCAPE) (Project ID: 875351; H2020-SC1-DTH-2019, SC1-DTH-01-2019; https://cordis.europa.eu/project/id/875351). This study was also supported by Fundación iSYS Internet Salud y Sociedad in the form of grants awarded to MA.F-E through the same ASCAPE project (Project ID: 875351), and grant Fundación "la Caixa" (LCF/PR/AR19/51450002) awarded to I.G-C and M.M-M. The funders had no role in study design, data collection and analysis, decision to publish, or preparation of the manuscript.

**Competing interests:** I have read the journal's policy and the authors of this manuscript have the following competing interests: Maria-Angeles Fuentes-Expósito and Imma Grau-Corral are affiliated with the iSYS Foundation, the organization responsible for developing and promoting the Xemio-Research App.

symptom tracking with conventional assessments may enhance personalized care and survivorship outcomes. This study is a sub-study of the clinical trial registered under ClinicalTrials.gov (Identifier: NCT05401643).

## Introduction

### Breast cancer survivors

Breast cancer is the most common cancer globally and significantly impacts cancer-related mortality rates [1,2]. However, advances in early detection and treatment have led to a steadily increasing population of cancer survivors, presenting both opportunities and challenges for oncology care. In 2020, approximately 5% of the population in Europe were living after a cancer diagnosis, equating to about 23.7 million people out of 477.9 million in the 29 European countries studied [3].

The long-term and late effects experienced by breast cancer survivors and the treatments they undergo entail considering various factors such as the type, duration, and dosage of treatment, as well as the patient's age during treatment. The assessment and management of these effects, such as quality of life, lymphedema, cardiotoxicity, cognitive impairment, distress, fatigue, bone health, musculoskeletal health, pain, neuropathy, infertility, sexual health, body image issues, and premature menopausal symptoms, among others, are necessary [4–6].

### mHealth and PROMs

Patient-Reported Outcome Measures (PROMs) collect self-reported information directly from patients about their health, quality of life, and functionality, offering insights unfiltered by healthcare providers. Mobile health (mHealth) technologies, such as smartphone apps and wearable devices, enhance this process by enabling remote monitoring, personalized interventions, and the seamless integration of PROMs and objective measures (OM) [7]. These tools empower breast cancer survivors to track symptoms, medication, and physical activity, fostering engagement in their care journey [8]. Notably, ICHOM's [9] standardized PROMs for breast cancer informed the Xemio app's development and symptom domains, addressing key physical, emotional, and social well-being aspects, including pain, fatigue, and body image. Integrating these measures into mHealth platforms enhances data-driven, personalized care, with studies highlighting the potential of such interventions to improve symptom management, self-efficacy, and communication with providers [10–14].

Despite these advancements, challenges remain in sustaining adherence and engagement with mHealth tools, as shown by research emphasizing the need for optimized strategies [15,16]. Apps like the Xemio-Research App, designed to track side effects, deliver reliable health information, and support breast cancer patients during treatment and recovery, exemplify the potential of mHealth in survivorship care. Our real-world investigation into Xemio's clinical integration aimed to assess

its impact on patient behavior and well-being, addressing survivors' informational and emotional needs while promoting a holistic, patient-centric approach to cancer care [17,18].

## Objectives

This study aimed to investigate the feasibility and acceptability of the mHealth tool Xemio in a clinical setting as a support tool for breast cancer survivors. This included analyzing user adherence, system usability assessed by the System Usability Scale [19] (SUS), and patient experience through an ad-hoc questionnaire.

The secondary objective was to examine the behavior of patients in reporting side effects, by comparing self-reported effects, using the Common Terminology Criteria for Adverse Events (CTCAE) [20] provided by the Xemio-Research App (52 effects), with those reported in traditional questionnaires. Both methods comply with the ICHOM standards for PROMs in breast cancer care.

## Materials and methods

This study is an experimental and prospective study, focused on evaluating specific outcomes in the intervention group where the Xemio-Research App was installed. The study was conducted within the framework of the European ASCAPE project.

### ASCAPE, OntoCR and Xemio-Research App

ASCAPE (Artificial Intelligence Supporting Cancer Patients across Europe) was a Horizon 2020 (H2020) project focused on breast and prostate cancer, two of the most prevalent malignancies [21]. Its primary goal was to use Artificial Intelligence and machine learning to improve the health and quality of life (QoL) of cancer patients through four pilot studies [22,23]. Clinical collaborators used validated questionnaires to assess various QoL aspects of these cancers. These questionnaires were integrated with data on daily activities, treatment side effects, and medical interventions, enabling Artificial Intelligence models to predict and recommend QoL improvements. The ASCAPE project collects HRQoL scores using validated instruments, including the EORTC QLQ-C30 [24], EORTC QLQ-BR23 [24], Hospital Anxiety and Depression Scale (HADS) [25] and Three-item Loneliness Scale (TILS) [26].

The Barcelona pilot of the ASCAPE project at Hospital Clínic de Barcelona (HCB) used OntoCR, an ontology-driven clinical repository [27,28] that received data from the Xemio-Research App and the clinical study questionnaires. By leveraging a dual model paradigm, OntoCR ensured seamless data transfer and semantic integrity using EN/ISO 13606-compliant extracts [18]. Xemio-Research (Fig 1) is an app developed in 2020 for breast cancer patients, providing them with proper information, enabling symptom tracking, and collecting physical activity data from its users daily (steps, time of activity, and calories). The deployment of the Xemio-Research backend was carried out on a server within the information systems area of the HCB [18].

### Ethical considerations

This study is part of the research approved by the Ethics Committee for Drug Research of the Hospital Clínic de Barcelona, Spain (HCB/ 2020/0971). All participants provided written informed consent in person during the recruitment visit, covering both primary and secondary analyses. The signed consent forms were securely stored at the hospital, with access restricted exclusively to the authorized project investigators. All data presented in this paper were anonymized. This study is a sub-study of the clinical trial registered under ClinicalTrials.gov Identifier: NCT05401643.

All data collected through the Xemio-Research app were pseudonymized and securely stored in compliance with the General Data Protection Regulation (GDPR). Access to the database was restricted to authorized investigators, and no

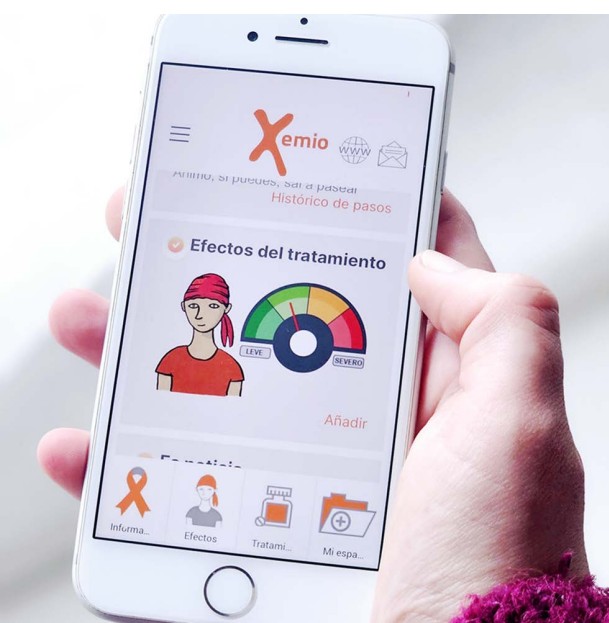

**Fig 1. Xemio App Interface.** The figure shows a user holding a smartphone displaying the Xemio app, developed by Fundación iSYS. Reprinted under a CC BY license, with permission from Fundación iSYS, original copyright 2024.

personally identifiable information was shared externally. These safeguards were implemented to ensure confidentiality and foster patient trust, which is a critical factor for the acceptance of digital health tools.

## Study design and participation

This one-year longitudinal feasibility study used a convenience sample of eligible breast cancer survivors recruited at a single academic center. The sample size was determined by the number of patients available and willing to participate during the recruitment window (December 1, 2020–December 22, 2021), with participants assigned to the Xemio intervention group. Recruitment depended on patient availability in the outpatient oncology department and was influenced by COVID-19–related constraints.

## Eligibility criteria

Patients were eligible for the study if they met the following criteria: they were breast cancer survivors without hospital treatment in the previous year, they owned a smartphone, they had no cancer-related complications, and they agreed to participate in the study.

## Procedures

Participation in the study was proposed to a consecutive sample of adult breast cancer survivors attending the Oncology outpatient consultations of the HCB who met the inclusion criteria. The recruitment period was 12 months, and the participation period in the study was an additional 12 months. For patients who agreed to participate in the clinical study and were assigned to the Xemio group, the App was installed. During a motivational interview, the functionality of the App was explained to them, and they were provided with a brochure containing information and a contact telephone number in case they encountered any issues. Patients were asked to make free use of the App and its contents. Throughout the

study, two reminder calls to use the App were given, coinciding with the collection of electronic patient-reported outcomes (ePROs).

## Outcome measures

Adherence to the application was measured using three available records: side effects, QoL questionnaires and number of daily steps (pedometer). Side effects were actively reported by the patients using the App's symptom tracker, which included 52 effects and their intensity according to CTCAE. Daily steps were passively obtained by the mobile phone's accelerometer. QoL questionnaires were completed at months 0, 3, 6, 9, and 12 according to ASCAPE design. Partial results from these questionnaires could prompt a call from the treating oncologist, who had access to the patients' follow-up data. At the end of the study, a satisfaction survey was conducted (see S1 and S2 Files). All the collected data was integrated into a pseudo-anonymized dataset to assess the overall feasibility and acceptability of the Xemio-Research mHealth tool in a clinical setting.

## Data analysis

The analysis focused on calculating means and standard deviations, providing a detailed summary of central tendency and variability to support the interpretation of the statistical findings.

   **Missing data.**  Given the feasibility design and attrition over time, analyses were conducted primarily as complete-case summaries with denominators clearly indicated for each time point and measure (e.g., number of respondents contributing to $T = 12$ questionnaire domains). No imputation was performed. This approach preserves transparency about data availability while avoiding assumptions that may not hold in a small convenience sample.

   **Adherence assessment.**  Adherence was assessed using two metrics: the frequency of side effect reports and the number of days with recorded step data. These indicators provide a detailed understanding of how consistently patients interacted with the app and the extent of their participation in the intervention.

   **Feasibility indicators.**  To evaluate the feasibility of the Xemio-Research App in a real-world setting, two questionnaires were employed. The first was the System Usability Scale (SUS) [19,29], which assessed perceived usability. The second was an ad-hoc questionnaire (S1 File) comprising four sections: (1) nine items derived from the Single Ease Question (SEQ) [30] to evaluate ease of use, (2) two open-ended questions soliciting qualitative feedback, (3) a scale assessing the frequency of use for different app functionalities, and (4) a final question asking participants whether they would recommend the app.

   **Comparing side effect capture: CTCAE vs. QoL questionnaires.**  We conducted a comparison between the spontaneously reported side effects collected using the CTCAE list and the standardized capture of symptoms through QoL questionnaires. Patients were encouraged to report side effects they experienced, choosing them from a list of 52 side effects presented in the app. The process was designed to be user-friendly and informative, ensuring that patients could accurately document their symptoms. Upon selecting a side effect, a screen appeared with a brief explanation. By clicking the next button, patients could view a list of intensities corresponding to those indicated by the CTCAE, with language adapted for patient understanding. After selecting the intensity (Fig 2), the screen displayed advice tailored to that intensity, providing immediate, relevant information and guidance based on the severity of the reported side effect. The side effect, its intensity, and the date of the report were recorded. Each side effect can have 1–4 intensity options (detailed in S2 File): 1 for mild symptoms, 2 for moderate, 3 for severe, and 4 for life-threatening symptoms.

## Results

Between December 2020 and November 2021, a total of 121 patients were recruited for the Barcelona ASCAPE Pilot. Of these, 61 patients enrolled in the Xemio-Research intervention. However, 12 patients withdrew from the study for reasons such as opting not to use the Xemio app during the study or experiencing a relapse that rendered them ineligible based

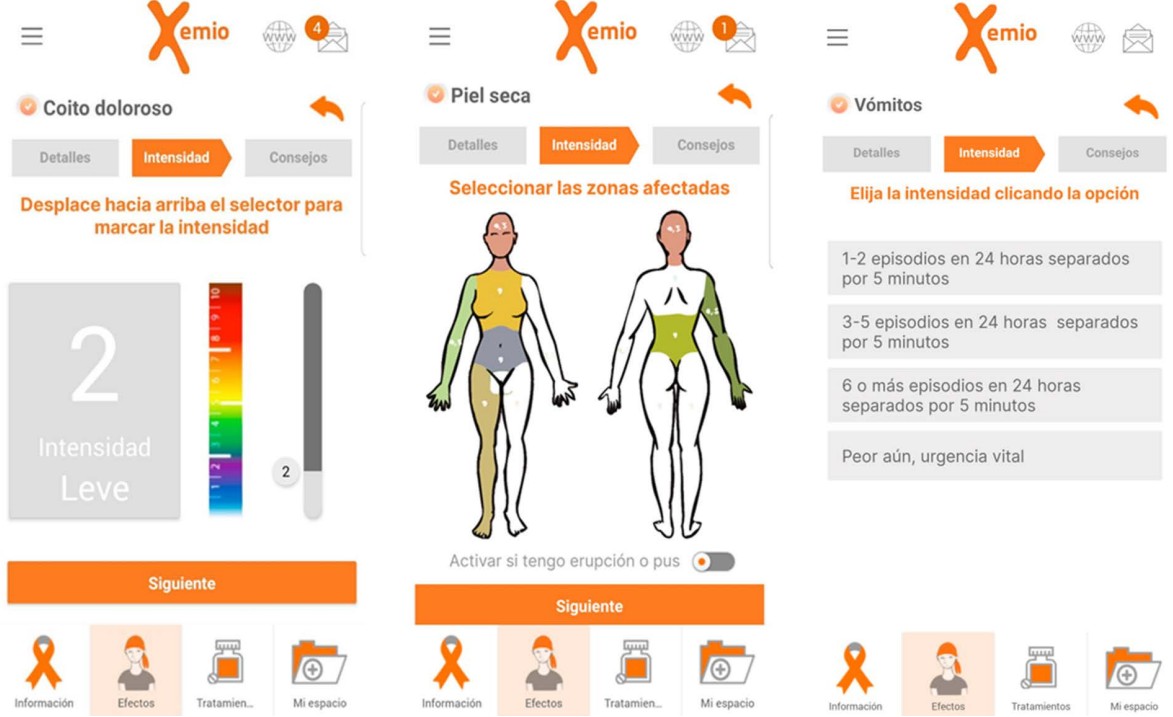

**Fig 2. Patient-reported side effect screens in the Xemio app.** Examples of the user interface for patient-reported symptom tracking in the Xemio mHealth app. (a) Reporting painful intercourse, where users can adjust the intensity level using a color-coded slider. (b) Reporting dry skin, where users select affected body areas and can indicate additional symptoms like rash or pus. (c) Reporting vomiting, where users choose intensity levels based on predefined episodic frequency categories. These features allow real-time symptom tracking, facilitating personalized patient management. Reprinted under a CC BY license, with permission from Fundación iSYS, original copyright 2020.

on the inclusion criteria. This left a final cohort of 49 patients who actively used the Xemio-Research App. The AI results to support treatment decisions developed by ASCAPE [23] didn't impact the present study, as its AI predictions were activated once the study was completed.

### Sociodemographic characteristics of the study sample

The participants in the sample had a mean age of 58.9 years. Regarding BMI, 38.3% had a normal weight, 36.2% were classified as overweight, and 25.5% were categorized as obese. In terms of employment, 40.4% were on labor incapacity, while 14.9% were unemployed. Most participants (80.6%) were undergoing treatment with aromatase inhibitors. Educational attainment varied, with 31.9% having no formal education and 29.8% holding university degrees. Economically, 40.4% resided in high-income districts, with an income ranging from €38.604 to €98.883. Additionally, 53.2% had never smoked, and 29.8% reported never consuming alcohol.

### Adherence results

Adherence to the Xemio-Research intervention was analyzed as the frequency of app usage throughout the study period. Fig 3 showed a significant decline in app usage after the first three months, with the highest adherence observed during the initial trimester. Usage dropped sharply in subsequent periods and stabilized at approximately 32% during the latter half of the year.

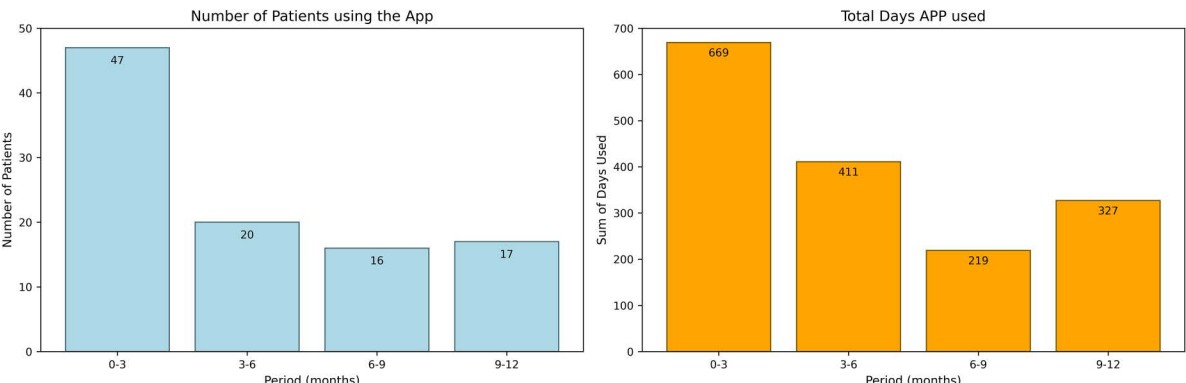

**Fig 3. App usage analysis.** Visualization of patient engagement with the Xemio app over a 12-month period. (a) The number of patients actively using the app in four time intervals: 0-3, 3-6, 6-9, and 9-12 months. A decline in user engagement is observed over time. (b) The total number of app usage days accumulated by patients in each time period, showing a similar declining trend. Despite a decrease in active users, the remaining users maintained relatively high engagement levels.

Fig 3 illustrates the usage of the app by patients at quarterly intervals throughout the year. During the study period, 49 patients used the Xemio-Research app. In the first 3 months, 47 patients (95.92%) accumulated 669 days of usage. By the 3–6 month period, usage dropped to 20 patients (40.82%) with 411 days. In the 6–9 months, 16 patients (32.65%) used the app for 219 days. Finally, in the 9–12 month period, 17 patients (34.69%) accumulated 327 days of usage. Despite the decrease in active users, the average days of use per patient increased in the later periods.

## Feasibility and acceptability

At the end of the study, participants completed an evaluation using the SUS questionnaire and the ad-hoc feasibility questionnaire (S1 File). The following sections present the results.

## SUS questionnaire results

Of the 49 patients who used the app, 36 completed the System Usability Scale (SUS) questionnaire. The SUS score, calculated from responses to the 10-item usability questionnaire, was 82.78 (SD = 14.21), indicating a level of usability classified as "excellent" [25–27].

## Ad-Hoc questionnaire results

A total of 33 patients completed the ad-hoc feasibility survey with four sections, designed to evaluate the perceived ease of use of each app functionality. The results from the first section of the questionnaire ad-hoc are detailed in Table 1, where 9 questions to answer using a Likert Scale ranging from 1 (Totally disagree) to 5 (Totally agree).

In the second part of the ad-hoc questionnaire, patients were asked to respond to the open-ended questions: *"What did you like most about the Xemio-Research App?"* and *"What did you like least about the Xemio-Research App?"* Participants provided qualitative feedback in free-text format.

1. Positive comments emphasized the app's ability to reassure users about normal symptoms, thereby reducing fear and anxiety.

**Table 1. Ad-hoc Xemio assessment.**

| Item | Question (Totally disagree 1–5 Totally agree) n = 33 | Mean (SD) |
|---|---|---|
| 1 | The application has been easy to use. | 4.35 (0.81) |
| 2 | It has been able to orient itself within the application. | 4.41 (0.86) |
| 3 | The application has allowed you to track your symptoms. | 3.72 (1.16) |
| 4 | It has been easy to assess the intensity of your symptoms. | 3.48 (1.02) |
| 5 | It has been easy to add treatments. | 3.72 (1.28) |
| 6 | It has been easy to find events. | 4.38 (1.04) |
| 7 | You have found events you are interested in? | 3.87 (1.2) |
| 8 | Did you find the information you consulted on the app useful? | 4.13 (1.13) |
| 9 | Has hygienic-dietary advice helped to alleviate your symptoms? | 3.62 (1.10) |
| | **TOTAL SCORE** | **3.96 (0.34)** |

2. Patients also highlighted its user-friendly interface, high-quality content, personalized symptom tracking, and the wide range of topics covered.

3. The app's visual appeal, ease of navigation, and accessibility to informative resources were frequently praised.

Conversely, participants identified areas for improvement, including:

1. smartphone compatibility issues,

2. difficulties in symptom tracking,

3. information overload,

4. technical problems,

5. perceived impersonality.

Some participants further explained that the perceived impersonality of the app made them feel it acted as a constant reminder of their illness.

In the third section of the ad-hoc questionnaire, patients evaluated the app's functionalities using a scale ranging from 1 to 5, reflecting their frequency of use. Among the 31 patients who completed this section, 21 provided ratings for usage frequency. As illustrated in Fig 4, the distribution of scores (1–5) across five app functionalities: Consult Information, Symptom Tracking, Medication Logging, Physical Activity, and Event Schedule.

Higher scores (4 and 5) are predominantly concentrated in the Consult Information and Symptom Tracking categories, reflecting more frequent use of these functionalities. Conversely, the other categories show a broader spread of scores, indicating greater variability in participant engagement with these features. In the fourth and final section of the ad-hoc questionnaire, patients were asked whether they would recommend the app. Of the 31 respondents, 27 (87.1%) answered affirmatively.

## Comparing side effect capture: CTCAE vs. QoL questionnaires

To evaluate the consistency and reliability of symptom reporting across different methods, we analyzed data from both patient-driven and standardized approaches. Symptoms recorded through the non-mandatory self-reporting feature of the Xemio-Research App were compared with those systematically collected via validated questionnaires (ASCAPE design).

Side effects were documented for patients who reported at least one symptom during the study year, using the CTCAE-based application list. These were then compared with side effects recorded through standard questionnaires. Data was

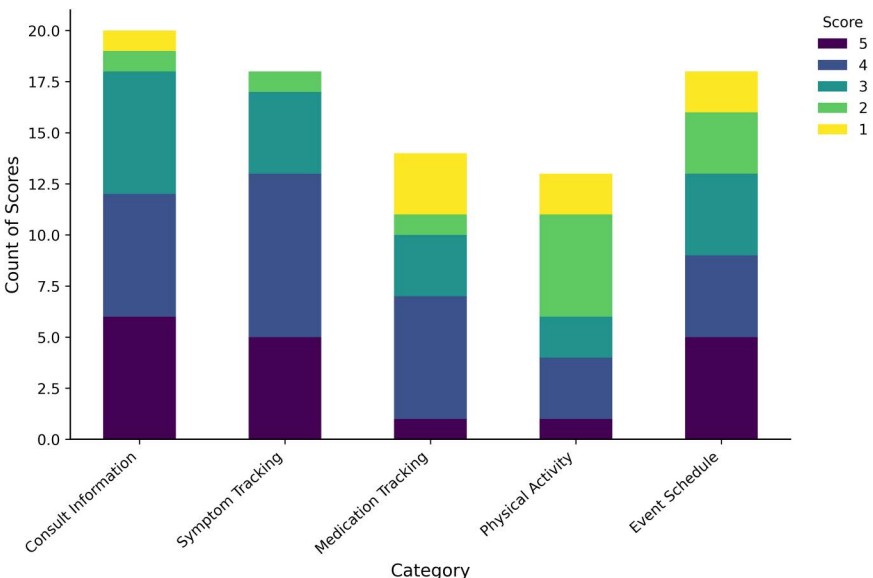

**Fig 4. Usage frequency of different app's functionalities by individual patients.** Stacked bar chart illustrating the frequency with which participants used five key features of the Xemio app: (a) Consult Information, (b) Symptom Tracking, (c) Medication Tracking, (d) Physical Activity, and (e) Event Schedule. Scores range from 1 (lowest usage) to 5 (highest usage), represented by different colours.

available for 39 of the 52 side effects included in the CTCAE-Xemio system, with varying intensity levels. The median intensity was classified as mild (≤1.4), moderate (1.5–2.4), severe (2.5–3.4), or life-threatening (>3.4).

The results are summarized in two tables, one with the CTCAE-reported effects, their median intensity, and frequency, comparing them to equivalent questionnaire items, and another that examines questionnaire responses and reported intensities, contrasting them with data from the app's symptom tracker to highlight similarities and differences between the two methods.

Table 2 provides an overview of side effects reported at least one time per patient, ranked by frequency, and compares them with data from the EORTC QLQ-C30, EORTC QLQ-BR23, and HADS questionnaires administered at T = 12 of the ASCAPE study. It identifies both exact matches and similar symptoms, offering a detailed comparison that underscores the similarities and distinctions between data collected via the app and traditional questionnaires.

The app provides a more detailed recording of side effects compared to standard questionnaires. CTCAE descriptions often differ substantially from those in validated QoL tools. For example, *pain* in the QLQ-C30 is broken down into more specific categories such as *joint pain*, *bone pain*, and *headache*, enabling the app to capture key distinctions. Side effects like *joint pain* (13 patients), *tingling* (10 patients), *muscle weakness* (9 patients), and *nail problems* (9 patients) are frequently reported in the app but are not specifically addressed in traditional questionnaires.

The intensity of symptoms reported in the Xemio-Research list of possible effects was generally rated by patients as high, with 69.23% (27/39) of symptoms rated as more than mild. Specific symptoms had mean intensity scores indicating moderate to severe levels of discomfort: *dyspareunia* (2.6), *joint pain* (2.3), *vaginal dryness* (2.2), and *insomnia* (2.0). *Anxiety* and *fatigue* had median scores of 1.8 and 1.7, respectively, indicating mild to moderate intensity. Other symptoms, such as *tingling*, *muscle weakness*, *muscle pain*, and *memory loss*, also had median scores of 1.7. *Hot flushes* had a median intensity score of 1.5, indicating a generally mild intensity.

Table 3 shows the median score for the EORTC QLQ-C30, EORTC QLQ-BR23, TILS, and HADS questionnaires, and the median intensity for the CTCAE. Some side effects reported in the CTCAE-Xemio app have a direct correspondence

**Table 2. Correspondence of CTCAE-Xemio Side Effects and QoL Questionnaire Items.** All questionnaires' results are from T = 12. [a] In the EORTC QLQ-C30 there is a generic section dedicated to pain. [b] In the EORTC QLQ-C30 Nausea/vomiting.

| Side-effect item (n) | CTCAE-Xemio intensity (median) | Reported symptom | QLQ Questionnaire | Scale value (median) | Type match |
|---|---|---|---|---|---|
| Joint pain (n = 13) | Moderate (2.4) | Pain[a] | QLQ-C30 | Low-Good (27.03) | Similar |
| Tingling or numbness (n = 10) | Moderate (1.7) | – | – | – | NA |
| Nail problems (n = 9) | Mild (1) | – | – | – | – |
| Muscle weakness (n = 9) | Moderate (1.7) | – | – | – | – |
| Anxiety (n = 8) | Moderate (1.8) | Anxiety | HADS | Normal (7.14) | Exact |
| Dry eye (n = 8) | Mild (1) | – | – | – | – |
| Hot flashes (n = 8) | Moderate (1.5) | – | – | – | – |
| Muscle pain (n = 7) | Moderate (1.7) | Pain[a] | QLQ-C30 | Low-Good (27.03) | Similar |
| Memory loss (n = 7) | Moderate (1.7) | Cognitive Functioning | QLQ-C30 | High-Good (77.03) | Similar |
| Insomnia (n = 6) | Moderate (2) | Insomnia | QLQ-C30 | Medium-Bad (38.74) | Exact |
| Vaginal dryness (n = 6) | Moderate (2.2) | Sexual Functioning | QLQ-BR23 | Low-Bad (16.67) | Similar |
| Dry mouth (n = 5) | Moderate (1.2) | – | – | – | – |
| Dyspareunia (n = 5) | Severe (2.6) | Sexual Enjoying | QLQ-BR23 | Middle-Bad (47.92) | Similar |
| Fatigue (n = 5) | Moderate (1.8) | Fatigue | QLQ-C30 | Medium-Bad (30.48) | Exact |
| Weight gain (n = 4) | Moderate (1.5) | – | – | – | – |
| Hair loss (n = 4) | Mild (1) | Hair Loss | QLQ-BR23 | Middle-Bad (62.41) | Exact |
| Abdominal pain (n = 3) | Moderate (1.7) | Pain[a] | QLQ-C30 | Low-Good (27.03) | Similar |
| Headache (n = 3) | Severe (2.7) | Pain[a] | QLQ-C30 | Low-Good (27.03) | Similar |
| Pain in the operated breast (n = 3) | Mild (1.3) | Pain[a], Breast Symptoms | QLQ-C30, QLQ-BR23 | Low-Good (27.03), Low-Good (17.79) | Similar |
| Chills (n = 3) | Mild (1.3) | – | – | – | – |
| Hypertension (n = 3) | Moderate (1.7) | – | – | – | – |
| Difficult digestion (n = 2) | Moderate (1.5) | – | – | – | – |
| Lymphedema (n = 2) | Mild (1) | Arm Symptoms | QLQ-BR23 | Low-Good (18.77) | Exact |
| Nausea (n = 2) | Moderate (1.5) | Nausea[b], Vomiting[b] | QLQ-C30 | Low-Good (4.50) | Similar |
| Diarrhea (n = 1) | Mild (1) | Diarrhea | QLQ-C30 | Low-Good (5.41) | Exact |
| Gum pain (n = 1) | Moderate (2) | Pain[a] | QLQ-C30 | Low-Good (27.03) | Similar |
| Toothache (n = 1) | Moderate (2) | Pain[a] | QLQ-C30 | Low-Good (27.03) | Similar |
| Constipation (n = 1) | Moderate (2) | Constipation | QLQ-C30 | Low-Good (18.92) | Exact |
| Weight loss (n = 1) | Moderate (2) | Appetite loss | QLQ-C30 | Low-Good (10.81) | Similar |
| Gastroesophageal reflux (n = 1) | Mild (1) | – | – | – | – |

in the questionnaires, while others are more specific in the app but generalized in the questionnaire. For example, the questionnaire uses *Pain* as a broad category, whereas the app distinguishes between *Headache* and *Abdominal pain*. The "Type match" column indicates whether the correspondence is exact or similar.

## Discussion

This study, conducted from November 2020 to November 2022, in the context of the ASCAPE project, evaluated the patient acceptance and usability of the Xemio-Research App among 49 participants. The sociodemographic and clinical profile of the study sample highlights a population predominantly of middle-aged women with diverse educational and economic backgrounds, a high prevalence of labour incapacity or unemployment, and a significant proportion undergoing aromatase inhibitor treatment.

**Table 3. Median results obtained by questionnaire dimension (T = 12).**

| Questionnaire's name (n) | Question' dimension | Scale value (median) | CTCAE-Xemio Name (n) | Intensity (median) | Type match |
|---|---|---|---|---|---|
| EORTC QLQ-C30 (n = 48) | Appetite Loss | Low-Good (10.81) | Anorexia (n = 2) | Moderate (2) | Similar |
| | Appetite Loss | Low-Good (10.81) | Weight loss (n = 1) | Moderate (2) | Similar |
| | Cognitive Functioning | High-Good (77.03) | Memory Loss (n = 7) | Moderate (1.7) | Similar |
| | Constipation | Low-Good (18.92) | Constipation (n = 1) | Mild (1) | Exact |
| | Diarrhea | Low-Good (5.41) | Diarrhea (n = 1) | Mild (1) | Exact |
| | Dyspnoea | Low-Good (18.02) | – | – | NA |
| | Emotional Functioning | Medium-Bad (67.12) | – | – | – |
| | Fatigue | Medium-Bad (30.48) | Fatigue (n = 5) | Moderate (1.8) | Exact |
| | Financial Problems | Low-Good (5.41) | – | – | – |
| | Global health status/QoL | High-Good (70.72) | – | – | – |
| | Insomnia | Medium-Bad (38.74) | Insomnia (n = 6) | Moderate (2) | Exact |
| | Nausea/Vomiting | Low-Good (4.50) | Nausea (n = 2) | Moderate (1.5) | Similar |
| | Pain | Low-Good (27.03) | Joint Pain (n = 13) | Moderate (2.4) | Similar |
| | Pain | Low-Good (27.03) | Muscle Pain (n = 7) | Moderate (1.7) | Similar |
| | Pain | Low-Good (27.03) | Abdominal Pain (n = 3) | Moderate (1.7) | Similar |
| | Pain | Low-Good (27.03) | Headache (n = 3) | Severe (2.7) | Similar |
| | Pain | Low-Good (27.03) | Pain in the operated breast (n = 3) | Mild (1.3) | Similar |
| | Pain | Low-Good (27.03) | Gum Pain (n = 1) | Moderate (2) | Similar |
| | Pain | Low-Good (27.03) | Bone Pain (n = 1) | Moderate (2) | Similar |
| | Pain | Low-Good (27.03) | Toothache (n = 1) | Moderate (2) | Similar |
| | Physical Functioning | High-Good (86.85) | – | – | – |
| | Role Functioning | High-Good (88.10) | – | – | – |
| | Social Functioning | High-Good (88.69) | – | – | – |
| EORTC QLQ-BR23 (n = 48) | Arm Symptoms | Low-Good (18.77) | Lymphedema (n = 2) | Mild (1) | Exact |
| | Body Image | High-Good (79.05) | – | – | – |
| | Breast Symptoms | Low-Good (17.79) | Pain in the operated breast (n = 3) | Mild (1.3) | Similar |
| | Future Perspective | Middle-Bad (58.56) | – | – | NA |
| | Hair Loss | Middle-Bad (62.41) | Hair Loss (n = 4) | Mild (1) | Exact |
| | Sexual Enjoyment | Middle-Bad (47.92) | Dyspareunia (n = 5) | Severe (2.6) | Similar |
| | Sexual Functioning | Low-Bad (16.67) | Vaginal dryness (n = 6) | Moderate (2.2) | Similar |
| | Systemic Therapy | Low-Good (24.71) | – | – | NA |
| TILS (n = 48) | Loneliness | Moderate (3.78) | – | – | NA |
| HADS (n = 48) | Anxiety | Normal (7.14) | Anxiety (n = 8) | Moderate (1.8) | Exact |
| | Depression | Normal (5.50) | Depression (n = 2) | Moderate (1.5) | Exact |

## Adherence to Xemio-Research Intervention

Adherence to the Xemio-Research App declined significantly over the study period. Initially, 95.92% of participants used the app in the first three months, but this dropped to 40.82% in the 3–6 month period, 32.65% in the 6–9 month period, and 34.69% in the 9–12 month period. Other studies [10] show that the average duration of mHealth interventions for breast cancer is 12 weeks. Similarly, German reimbursement for digital health applications (DiGA) is set at 30–90 days, showing some alignment with the behavior of users of the Xemio-Research App. Despite the drop in users, those who continued using the app maintained high engagement. This trend is common in digital health interventions [15], where initial enthusiasm wanes, highlighting the need for strategies to sustain long-term engagement, such as: develop

gamification features allowing users to participate in daily or weekly challenges that positively reinforce their commitment and motivation and incorporate more effective tracking and real-time monitoring, which could be key to improving adherence and the effectiveness of digital health interventions.

**Interpreting the decline in adherence**

Beyond the typical waning of engagement observed in digital health tools, several factors may have contributed to the observed decline. First, *digital literacy and device heterogeneity* (reported by participants as compatibility and usability issues) can create friction that accumulates over time, particularly in older survivors. Second, *socioeconomic constraints* (e.g., unemployment or labor incapacity) and *treatment-related fatigue* may reduce the cognitive and emotional bandwidth available for sustained self-tracking. Third, *psychological factors* such as anxiety or information overload may lead some users to avoid repeated symptom logging if it amplifies illness salience. These mechanisms are consistent with prior reviews on engagement with mHealth in cancer care and eHealth adherence, which highlight usability, burden, and motivational fit as key drivers of sustained use [7,8,11,15,16].

**Implications for clinical adoption and sustainability**

To enhance long-term use in survivorship care, interventions should (i) simplify key flows for symptom logging and reduce optional complexity; (ii) calibrate *just-in-time* prompts to patient context to avoid alert fatigue; (iii) offer *lightweight* motivational features (e.g., streaks or progress check-ins) tied to clinically meaningful behaviors rather than points alone; and (iv) integrate *closed-loop* feedback (e.g., clinician-acknowledged summaries or tailored advice) that reinforces perceived clinical value. Real-time CTCAE tracking may be most sustainable when embedded in routine follow-up workflows with *minimal patient effort* and *visible clinical payoff*.

**Usability and participant feedback**

The SUS results indicated excellent usability, with an average score of 82.78, suggesting that participants found the app user-friendly and well-integrated. The ad-hoc feasibility questionnaire reinforced these findings, with participants rating the app's ease of use and symptom-tracking features highly. The mean score for ease of use was 4.35, and 87.1% of respondents would recommend the app, reflecting high satisfaction levels. However, there were some areas for improvement. Functionalities preference varied, with symptom monitoring and consulting information being the most frequently used features.

**Symptoms and quality of life reporting**

In the validated questionnaires, participants reported a good overall quality of life, with high levels of physical and social functioning. Moderate challenges were observed in areas such as fatigue, insomnia, pain, and systemic therapy side effects, while emotional well-being was generally favourable, with normal levels of depression and mild anxiety. However, notable concerns included sexual functioning and cognitive issues, reflecting the broader impact of treatment.

The comparison between self-reported symptoms in the app and those captured via validated questionnaires reveals important differences. While tools like EORTC QLQ-C30, BR23, and HADS effectively capture patient-reported outcomes across broad domains, they may overlook specific symptoms. The CTCAE complements these tools by allowing precise, real-time tracking of specific symptoms, such as *joint pain* or *tingling*, that are not captured in traditional instruments. This integration can enhance the understanding of patient experiences and improve symptom management by bridging the gap between broad assessments and detailed symptom tracking.

Compared with traditional PROMs, the CTCAE-based tracking in the Xemio app provided finer granularity and timeliness. A concise comparison of strengths and limitations is summarized in Table 4. This approach facilitates clinical

**Table 4. Comparative strengths and limitations of PROMs and CTCAE-based symptom tracking.**

| Tool | Strengths | Limitations |
|---|---|---|
| PROMs | Validated, widely used, standardized tools that assess quality of life and psychological well-being across multiple domains (physical, emotional, social). Useful for research and longitudinal comparisons. | Limited sensitivity to specific or emerging symptoms (e.g., tingling, nail problems, joint pain); rely on retrospective recall, which may bias responses; periodic collection may delay clinical action. |
| CTCAE-based tracking | Real-time, granular monitoring of patient-prioritized toxicities; provides immediate feedback and clinically actionable data for oncologists; enables integration into routine follow-up workflows. | Requires sustained patient adherence; potential underreporting without prompts; focuses on physical toxicities and does not systematically capture broader psychosocial outcomes. |

integration by capturing clinically relevant toxicities as they occur, offering oncologists more actionable data than retrospective questionnaires.

These differences may also stem from psychological variations in how patients focus their attention. Traditional questionnaires operate through a "pull mechanism", systematically prompting patients to recall or recognize symptoms they might not spontaneously report. In contrast, the Xemio-Research App employs a "push mechanism", allowing patients to independently log symptoms in real time, influenced by their momentary focus and perception of symptom severity [31,32]. Combining these approaches provides a more holistic perspective, enabling comprehensive symptom reviews while capturing patient-prioritized outcomes. This dual strategy broadens the scope of symptom tracking, addressing both immediate and systemic concerns, and supports a nuanced understanding of patient well-being and treatment impact [33].

## Limitations

This study was conducted within the framework of the European artificial intelligence project ASCAPE. Additionally, the recruitment of patients was significantly impacted by the COVID-19 pandemic, leading to a reduced sample size and limited variability within the cluster. To mitigate disruptions, participants were offered remote support, including periodic telephone reminders and guidance on app use. Although these measures partially compensated for reduced in-person interaction, they could not fully offset the challenges posed by the pandemic. These constraints may affect the generalizability of the findings and highlight the need for further research under more stable conditions to validate and expand upon these results.

The study was based on a small, convenience sample of breast cancer survivors recruited at a single center, which limits generalizability. Moreover, the analysis relied primarily on descriptive statistics without formal inferential testing, reflecting the exploratory and feasibility-oriented design of this work.

These findings should be interpreted as exploratory and feasibility-focused. While they provide preliminary evidence supporting the usability and potential value of the Xemio-Research app, further studies with larger and more diverse samples are required to validate these results.

## Conclusion

The Barcelona ASCAPE Pilot suggests that the Xemio-Research App has the potential to support breast cancer survivorship care. While initial engagement was high and usability was strong, maintaining long-term adherence remains challenging. Participant feedback and symptom reports underscored the importance of continuous refinement and personalized support to enhance the effectiveness of digital health interventions. Future research should focus on these strategies to sustain engagement and further integrate real-time patient-reported outcomes with standardized clinical assessments to provide a comprehensive view of patient well-being.

## Supporting information

**S1 File. Xemio App Evaluation Ad-hoc questionnaire.**
(DOCX)

**S2 File. Xemio App' side effects measures by oncological categories CTCAE.**
(DOCX)

## Author contributions

**Conceptualization:** Maria-Angeles Fuentes-Expósito, Santiago Frid, Montserrat Muñoz-Mateu, Inmaculada Grau-Corral.

**Data curation:** Maria-Angeles Fuentes-Expósito, Santiago Frid.

**Formal analysis:** Santiago Frid, Montserrat Muñoz-Mateu, Antoni Sisó-Almirall, Inmaculada Grau-Corral.

**Funding acquisition:** Montserrat Muñoz-Mateu, Inmaculada Grau-Corral.

**Investigation:** Maria-Angeles Fuentes-Expósito, Santiago Frid, Montserrat Muñoz-Mateu, Inmaculada Grau-Corral.

**Methodology:** Maria-Angeles Fuentes-Expósito, Santiago Frid, Montserrat Muñoz-Mateu, Antoni Sisó-Almirall, Manuel Armayones Ruiz, Inmaculada Grau-Corral.

**Project administration:** Inmaculada Grau-Corral.

**Resources:** Maria-Angeles Fuentes-Expósito, Santiago Frid, Inmaculada Grau-Corral.

**Software:** Maria-Angeles Fuentes-Expósito, Santiago Frid.

**Supervision:** Santiago Frid, Montserrat Muñoz-Mateu, Manuel Armayones Ruiz, Inmaculada Grau-Corral.

**Validation:** Antoni Sisó-Almirall, Manuel Armayones Ruiz, Inmaculada Grau-Corral.

**Visualization:** Maria-Angeles Fuentes-Expósito.

**Writing – original draft:** Maria-Angeles Fuentes-Expósito, Santiago Frid, Inmaculada Grau-Corral.

**Writing – review & editing:** Maria-Angeles Fuentes-Expósito, Santiago Frid, Montserrat Muñoz-Mateu, Antoni Sisó-Almirall, Manuel Armayones Ruiz.

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
