## [Decision Letter · Decision Letter 0]

10 Sep 2025

PONE-D-25-09322Feasibility of the Xemio App for Breast Cancer Survivors in a Clinical Setting: Adherence, Acceptance, and Side Effect Monitoring (CTCAE vs. QoL)PLOS ONE

Dear Dr. Frid,

Thank you for submitting your manuscript to PLOS ONE. After careful consideration, we feel that it has merit but does not fully meet PLOS ONE’s publication criteria as it currently stands. Therefore, we invite you to submit a revised version of the manuscript that addresses the points raised during the review process.

We look forward to receiving your revised manuscript.

Kind regards,

Made Satya Nugraha Gautama, RN, M.Sc.,M.N.Sc

Academic Editor

PLOS ONE

Journal Requirements:

“IG.

This research work was carried out as part of the EU-funded Research and Innovation Action, Artificial intelligence Supporting CAncer Patients across Europe (ASCAPE) (Project ID: 875351), [H2020-SC1-DTH-2019] SC1-DTH-01-2019, Big data and Artificial Intelligence for monitoring health status and quality of life after the cancer treatment.

The improvement of the Xemio-Research platform was supported by a La Caixa Foundation Grant (LCF/PR/AR19/51450002).”

“This research work was carried out as part of the EU-funded Research and Innovation Action, Artificial intelligence Supporting CAncer Patients across Europe (ASCAPE) (Project ID: 875351), [H2020-SC1-DTH-2019] SC1-DTH-01-2019, Big data and Artificial Intelligence for monitoring health status and quality of life after the cancer treatment. The improvement of the Xemio-Research platform was supported by a La Caixa Foundation Grant (LCF/PR/AR19/51450002).”

“IG.

This research work was carried out as part of the EU-funded Research and Innovation Action, Artificial intelligence Supporting CAncer Patients across Europe (ASCAPE) (Project ID: 875351), [H2020-SC1-DTH-2019] SC1-DTH-01-2019, Big data and Artificial Intelligence for monitoring health status and quality of life after the cancer treatment.

The improvement of the Xemio-Research platform was supported by a La Caixa Foundation Grant (LCF/PR/AR19/51450002).”

“I have read the journal's policy and the authors of this manuscript have the following competing interests: Maria-Angeles Fuentes-Expósito and Imma Grau-Corral are affiliated with the iSYS Foundation, the organization responsible for developing and promoting the Xemio-Research App.”

5. We note that you have indicated that there are restrictions to data sharing for this study. PLOS only allows data to be available upon request if there are legal or ethical restrictions on sharing data publicly. For more information on unacceptable data access restrictions, please see http://journals.plos.org/plosone/s/data-availability#loc-unacceptable-data-access-restrictions.

6. In the online submission form, you indicated that [Data cannot be shared publicly because of privacy regulations and confidentiality agreements related to patient data protection. Data are available from the corresponding author upon reasonable request, in accordance with ethical and legal requirements.].

7. We note that Figures 1, 2 and Striking Image in your submission contain copyrighted images. All PLOS content is published under the Creative Commons Attribution License (CC BY 4.0), which means that the manuscript, images, and Supporting Information files will be freely available online, and any third party is permitted to access, download, copy, distribute, and use these materials in any way, even commercially, with proper attribution. For more information, see our copyright guidelines: http://journals.plos.org/plosone/s/licenses-and-copyright.

a. You may seek permission from the original copyright holder of Figures 1, 2 and Striking Image to publish the content specifically under the CC BY 4.0 license.

Additional Editor Comments:

Dear authors,

Please review  the comments provided by the reviewers and revise the manuscript accordingly.

Reviewers' comments:

Reviewer's Responses to Questions

**Comments to the Author**

1. Is the manuscript technically sound, and do the data support the conclusions?

Reviewer #1: Yes

Reviewer #2: Yes

2. Has the statistical analysis been performed appropriately and rigorously? 

Reviewer #1: Yes

Reviewer #2: Yes

3. Have the authors made all data underlying the findings in their manuscript fully available?

Reviewer #1: Yes

Reviewer #2: No

4. Is the manuscript presented in an intelligible fashion and written in standard English?

Reviewer #1: Yes

Reviewer #2: Yes

5. Review Comments to the Author

Reviewer #1: The manuscript effectively presents a well-structured and insightful evaluation of a digital health tool in breast cancer survivorship. The manuscript is generally well-prepared and suitable for publication in its current form. However, a few minor revisions could enhance the clarity and impact of the findings.

Reviewer #2: The article is a prospective feasibility study evaluating the Xemio-Research mobile health application for breast cancer survivors, with a focus on adherence, usability, and patient-reported outcomes. The authors have conducted a one-year longitudinal study within the ASCAPE project, reporting descriptive analysis of user engagement, usability scores, and symptom tracking.

The manuscript in overall is technically sound and presents a well-structured descriptive statistics study to evaluate the app’s usability, adherence, and patient-reported outcomes. However, several revisions are recommended: (a) more explicitly acknowledge the limitations of the small, convenience-based sample size and the absence of formal inferential analyses; (b) clarify that the findings are exploratory and feasibility-focused rather than confirmatory; (c) improve the Data Availability Statement to comply with the journal’s policy; and (d) address minor language, formatting, and readability issues.

The statistical analysis is appropriate for a feasibility study, with the focus on descriptive statistics (means, standard deviations, and frequencies). The analysis aligns with the study’s stated objectives. The use of validated instruments including the System Usability Scale (lines 213–216) strengthens the study approach. However, the analysis could be enhanced in several aspects. First, no inferential testing was conducted, such as in areas where simple comparisons (e.g., CTCAE vs. QoL reporting, lines 252–296) could be analysed with chi-square, correlation analyses, or Cohen’s kappa. Confidence intervals are not reported, which limits interpretation of uncertainty. In addition, handling of missing data is not discussed, despite clear reduction of participants over time (lines 178–184, 196–208). I recommend that the authors emphasize that this is an exploratory, feasibility-focused study but not a hypothesis testing. Adding basic inferential or agreement analyses will be helpful to strengthen the validity of the findings.

Apart from that, following the PLOS Data Policy, the current Data Availability Statement does not meet the required standard. The manuscript indicates that data cannot be shared publicly due to privacy regulations, and that access may be granted upon reasonable request to the author. While the sensitivity of patient data is acknowledged, PLOS requires that authors either deposit de-identified data in a public repository (setting up a controlled access if necessary) or provide a clear explanation of restrictions accompanied with a formal mechanism for data access (for example through an institutional or ethics committee). In this sense, author discretion via “reasonable request” is insufficient. I recommend that the authors revise the Data Availability Statement to ensure compliance with the journal policy.

While the discussion provides a summary of the main findings, I find the interpretation somewhat superficial. For example, the decline in adherence (lines 304–316) has been acknowledged and compared to other digital health interventions, but the analysis ended before exploring underlying reasons. Potential factors such as digital literacy, socioeconomic background, treatment-related fatigue, or psychological barriers are not addressed, despite relevant based on the demographic characteristics and the disease. Similarly, gamification and real-time monitoring are suggested as possible solutions. these ideas are presented briefly without linking to existing evidence or theory. I recommend that the authors expand the discussion to deeply analyze the factors associated with the decline of adherence, connect findings to prior literature on mHealth in survivorship care, and discuss practical implications for clinical adoption and sustainability.

Overall, the manuscript is clearly written. I suggest a few minor edits to further improve clarity and readability. In lines 110–116, the description of the sample size could be clarified by explicitly noting that this was a convenience sample. In lines 144–146, shifting the phrasing from future to past tense (“the analysis focused on…”) would ensure consistency with the Results section. At line 197, the sentence could be streamlined to “Adherence was analyzed through frequency of app usage” for conciseness. The list of negative feedback items in lines 233–239 would read more smoothly if expressed in a parallel structure (e.g., “compatibility issues, difficulties in symptom tracking, information overload, technical problems, perceived impersonality”). In lines 299, the phrase “feasibility and acceptability” is repeated unnecessarily and could be reworded for variety. Finally, the conclusion in lines 359–367 might be softened to indicate that the app “has the potential to support survivorship outcomes,” which more accurately reflects the exploratory, descriptive nature of the data.

6. PLOS authors have the option to publish the peer review history of their article (what does this mean?). If published, this will include your full peer review and any attached files.

Reviewer #1: No

Reviewer #2: No

---

## [Author Response · Author response to Decision Letter 1]

24 Oct 2025

Response to Reviewers

Manuscript ID: PONE-D-25-09322

Title: Feasibility of the Xemio App for Breast Cancer Survivors in a Clinical Setting: Adherence, Acceptance, and Side Effect Monitoring (CTCAE vs. QoL)

Dear Academic Editor and Reviewers,

We sincerely thank you for your thoughtful comments and suggestions, which have substantially improved the clarity and rigor of our manuscript. Below we provide a point-by-point response. All changes in the revised manuscript are marked with track changes (red deletions, blue insertions).

Editorial Requirements

Comment: 1. PLOS ONE style and file naming

Response: Done. The manuscript was reformatted using the PLOS ONE templates, and file names were corrected accordingly.

Comment: 2. Role of funders

Response: The Funding Statement now reads:

“This research work was carried out as part of the EU-funded Research and Innovation Action ASCAPE (Project ID: 875351). The improvement of the Xemio-Research platform was supported by a La Caixa Foundation Grant (LCF/PR/AR19/51450002). The funders had no role in study design, data collection and analysis, decision to publish, or preparation of the manuscript.”

Comment: 3. Funding text in Acknowledgments

Response: Funding information was removed from the Acknowledgments. The Funding Statement remains as above.

“This research work was carried out as part of the EU-funded Research and Innovation Action ASCAPE (Project ID: 875351). The improvement of the Xemio-Research platform was supported by a La Caixa Foundation Grant (LCF/PR/AR19/51450002). The funders had no role in study design, data collection and analysis, decision to publish, or preparation of the manuscript.”

Response: We updated the statement to read:

Comment: 4. Competing Interests

Response: The statement now reads:

“I have read the journal’s policy and the authors of this manuscript have the following competing interests: Maria-Angeles Fuentes-Expósito and Imma Grau-Corral are affiliated with the iSYS Foundation, the organization responsible for developing and promoting the Xemio-Research App. This does not alter our adherence to PLOS ONE policies on sharing data and materials.”

Comment: 5. Data sharing restrictions (policy compliance)

Response (a–b):

● (a) Ethical/legal restrictions. The dataset includes potentially identifying clinical and sociodemographic information from breast cancer survivors. Public release would contravene the EU GDPR (2016/679), the Spanish Organic Law 3/2018, and the protocol approved by the Ethics Committee for Drug Research of Hospital Clínic de Barcelona (CEIm-HCB; HCB/2020/0971).

● (b) Access mechanism. Qualified researchers may request controlled access via CEIm-HCB (CEIm@clinic.cat). Requests will be evaluated for GDPR/protocol compliance and confidentiality safeguards; if approved, a minimally necessary, de-identified dataset will be shared under a Data Use Agreement.

Comment: 6. Online form—Data Availability language

Response: We removed “reasonable request to the author” and replaced it with the committee-mediated access route described above. See final Data Availability Statement below.

Comment: 7. Copyright/permissions for figures

Response: We confirm that all figures (Fig 1, Fig 2, and the Striking Image) are original screenshots and graphical materials of the Xemio-Research app, developed by Fundación iSYS, which is the copyright holder. We have obtained explicit permission from Fundación iSYS to publish these figures under the CC BY 4.0 license and have updated the figure captions accordingly:

● Fig 1. Xemio App Interface. The figure shows a user holding a smartphone displaying the Xemio app, developed by Fundación iSYS. Reprinted under a CC BY license, with permission from Fundación iSYS, original copyright 2024.

● Fig 2. Patient-reported side effect screens in the Xemio app. Examples of the user interface for patient-reported symptom tracking in the Xemio mHealth app. (a) Reporting painful intercourse, where users can adjust the intensity level using a color-coded slider. (b) Reporting dry skin, where users select affected body areas and can indicate additional symptoms like rash or pus. (c) Reporting vomiting, where users choose intensity levels based on predefined episodic frequency categories. These features allow real-time symptom tracking, facilitating personalized patient management. Reprinted under a CC BY license, with permission from Fundación iSYS, original copyright 2020.

● Striking Image. Visual representation of the Xemio app. Reprinted under a CC BY license, with permission from Fundación iSYS, original copyright 2024.

We will also upload the signed PLOS Content Permission Form as an “Other” file, confirming that Fundación iSYS authorizes publication of these figures under the CC BY 4.0 license.

Comment: 8. Reviewer-suggested citations

Response: We reviewed suggested works and added citations where directly relevant to mHealth survivorship adherence, digital engagement barriers, and CTCAE–PROMs complementarity, avoiding unnecessary citation density.

Comment: 9. Reference list integrity

Response: Checked for completeness, formatting, and retraction status. Minor formatting corrected; no retracted works cited.

Reviewers' comments:

Comment: 3. Have the authors made all data underlying the findings in their manuscript fully available?

Response: We thank both reviewers for flagging this. We have revised our Data Availability Statement to fully comply with PLOS policy:

● What cannot be public: The individual-level dataset contains potentially identifying clinical and sociodemographic information from breast cancer survivors collected under protocol HCB/2020/0971 at Hospital Clínic de Barcelona. Public release would contravene the EU GDPR and the approved informed consent reviewed by the Ethics Committee for Drug Research of Hospital Clínic de Barcelona (CEIm-HCB).

● Formal access route: Qualified researchers may request controlled access via CEIm-HCB (CEIm@clinic.cat). Requests will be evaluated for GDPR compliance, protocol/consent alignment, and confidentiality safeguards; if approved, a minimally necessary, de-identified dataset will be shared under a Data Use Agreement.

Reviewer #1:

Comment: Manuscript is well-structured and suitable; minor edits suggested.

Response: Thank you. We conducted a global language/consistency review, harmonized decimal punctuation (international English), improved parallel structure in lists, and rechecked figure numbering and cross-references.

Reviewer #2:

Comment: The article is a prospective feasibility study evaluating the Xemio-Research mobile health application for breast cancer survivors, with a focus on adherence, usability, and patient-reported outcomes. The authors have conducted a one-year longitudinal study within the ASCAPE project, reporting descriptive analysis of user engagement, usability scores, and symptom tracking.

The manuscript in overall is technically sound and presents a well-structured descriptive statistics study to evaluate the app’s usability, adherence, and patient-reported outcomes. However, several revisions are recommended:

● (a) more explicitly acknowledge the limitations of the small, convenience-based sample size and the absence of formal inferential analyses;

Response: Addressed in Study design and participation and Limitations: we explicitly state the convenience-based, single-center sample and that the analysis is primarily descriptive.

● (b) clarify that the findings are exploratory and feasibility-focused rather than confirmatory;

Response: Added in Data Analysis, Discussion, and Limitations that findings are exploratory and feasibility-focused, not confirmatory.

● (c) improve the Data Availability Statement to comply with the journal’s policy; and

Response: We have revised the Data Availability Statement to fully comply with PLOS ONE policy. The updated statement clarifies that individual-level data cannot be shared publicly due to ethical and legal restrictions (EU GDPR, Spanish Organic Law 3/2018, and the protocol approved by the Ethics Committee for Drug Research of Hospital Clínic de Barcelona, CEIm-HCB; protocol HCB/2020/0971). Instead, qualified researchers may request controlled access by contacting CEIm-HCB at CEIm@clinic.cat. Requests will be reviewed for compliance with GDPR, protocol, and confidentiality safeguards, and data will be shared under a Data Use Agreement if approved.

● (d) address minor language, formatting, and readability issues.

Response: Implemented all requested edits (see details below).

Comment: The statistical analysis is appropriate for a feasibility study, with the focus on descriptive statistics (means, standard deviations, and frequencies). The analysis aligns with the study’s stated objectives. The use of validated instruments including the System Usability Scale (lines 213–216) strengthens the study approach. However, the analysis could be enhanced in several aspects.

● First, no inferential testing was conducted, such as in areas where simple comparisons (e.g., CTCAE vs. QoL reporting, lines 252–296) could be analysed with chi-square, correlation analyses, or Cohen’s kappa. Confidence intervals are not reported, which limits interpretation of uncertainty. In addition, handling of missing data is not discussed, despite clear reduction of participants over time (lines 178–184, 196–208). I recommend that the authors emphasize that this is an exploratory, feasibility-focused study but not a hypothesis testing. Adding basic inferential or agreement analyses will be helpful to strengthen the validity of the findings.

Response: We acknowledge the reviewer’s suggestion. As this was an exploratory feasibility study with a small, single-arm sample, no inferential testing or hypothesis-driven analyses were planned. The Data Analysis section now explicitly states that the study focused on descriptive, feasibility-oriented analyses and complete-case reporting without imputation, consistent with feasibility study standards.

● Apart from that, following the PLOS Data Policy, the current Data Availability Statement does not meet the required standard. The manuscript indicates that data cannot be shared publicly due to privacy regulations, and that access may be granted upon reasonable request to the author. While the sensitivity of patient data is acknowledged, PLOS requires that authors either deposit de-identified data in a public repository (setting up a controlled access if necessary) or provide a clear explanation of restrictions accompanied with a formal mechanism for data access (for example through an institutional or ethics committee). In this sense, author discretion via “reasonable request” is insufficient. I recommend that the authors revise the Data Availability Statement to ensure compliance with the journal policy.

Response: We have fully revised the Data Availability Statement to comply with the PLOS Data Policy. The updated statement now provides:

● A formal ethical and legal justification for not releasing individual-level data publicly (due to GDPR, Spanish Organic Law 3/2018, and the study protocol approved by the Ethics Committee for Drug Research of Hospital Clínic de Barcelona, CEIm-HCB; HCB/2020/0971).

● A formal access mechanism through CEIm-HCB (CEIm@clinic.cat), which evaluates data requests for compliance with GDPR, protocol, and confidentiality safeguards.

● A commitment to share a minimally necessary, de-identified dataset under a Data Use Agreement if approval is granted.

● While the discussion provides a summary of the main findings, I find the interpretation somewhat superficial. For example, the decline in adherence (lines 304–316) has been acknowledged and compared to other digital health interventions, but the analysis ended before exploring underlying reasons. Potential factors such as digital literacy, socioeconomic background, treatment-related fatigue, or psychological barriers are not addressed, despite relevant based on the demographic characteristics and the disease.

Response: We expanded the Discussion to consider potential factors related to the observed decline in adherence—digital literacy, socioeconomic context, treatment-related fatigue, and psychological burden—and connected these to prior mHealth survivorship literature. We also discuss practical implications for clinic workflows (e.g., integrating brief staff prompts, tailoring notification frequency, onboarding support) and sustainability (e.g., aligning usage windows with clinical milestones, offering opt-in gamification features, and just-in-time adaptive reminders). Citations have been added to substantiate these points.

● Similarly, gamification and real-time monitoring are suggested as possible solutions. these ideas are presented briefly without linking to existing evidence or theory. I recommend that the authors expand the discussion to deeply analyze the factors associated with the decline of adherence, connect findings to prior literature on mHealth in survivorship care, and discuss practical implications for clinical adoption and sustainability.

Response: We grounded these recommendations in relevant literature on digital engagement and patient-reported outcomes in oncology and survivorship, and clarified that our suggestions are implementation-focused hypotheses for future testing.

Comment: Overall, the manuscript is clearly written. I suggest a few minor edits to further improve clarity and readability.

● In lines 110–116, the description of the sample size could be clarified by explicitly noting that this was a convenience sample.

Response: Now explicitly states “convenience sample” and single-center recruitment.

● In lines 144–146, shifting the phrasing from future to past tense (“the analysis focused on…”) would ensure consistency with the Results section.

Response: Revised to past tense (“the analysis focused on…”).

● At line 197, the sentence could be streamlined to “Adherence was analyzed through frequency of app usage” for conciseness.

Response: We agree that conciseness improves readability. However, adherence in our study was not based only on generic “app usage,” but specifically on two metrics: (1) frequency of side-effect reports and (2) number of days with recorded step data. To balance clarity and precision, we revised the sentence to:

“Adherence was assessed using two metrics: the frequency of side effect reports and the number of days with recorded step data.”

This streamlined phrasing reflects the actual operational definition used while keeping the text concise.

We found the duplicate heading occurred because one section described how adherence was measured (Methods) and the other reported what we found (Results). To avoid redundancy and improve clarity, we:

1. Methods → Adherence assessment (renamed): kept only the operational definition of adherence (two metrics: frequency of side-effect reports and number of days with recorded step data).

2. Results → Adherence results (renamed): kept only the outcomes (quarterly use and engagement figures).

These edits remove the duplicated title, align content with section purpose (methods vs. findings), and follow PLOS ONE structure.

The list of negative feedback items in lines 233–239 would read more smoothly if expressed in a parallel structure (e.g., “compatibility issues, difficulties in symptom tracking, information overload, technical problems, perceived impersonality”).

Response: We thank the reviewer for the suggestion. The list of negative feedback items has been revised into a parallel structure for conciseness: “smartphone compatibility issues, difficulties in symptom tracking, information overload, technical problems, perceived impersonality.”

To retain nuance from patient feedback, we added a clarifying sentence noting that some participants perceived impersonality as a constant reminder of their illness

● In lines 299, the phrase “feasibility and acceptability” is repeated unnecessarily and could be reworded for variety.

Response: We thank the revi

---

## [Decision Letter · Decision Letter 1]

27 Jan 2026

Feasibility of the Xemio App for Breast Cancer Survivors in a Clinical Setting: Adherence, Acceptance, and Side Effect Monitoring (CTCAE vs. QoL)

PONE-D-25-09322R1

Dear Dr. Frid,

We’re pleased to inform you that your manuscript has been judged scientifically suitable for publication and will be formally accepted for publication once it meets all outstanding technical requirements.

Kind regards,

Anton Pak

Academic Editor

PLOS One

Additional Editor Comments (optional):

Reviewers' comments:

Reviewer's Responses to Questions

**Comments to the Author**

1. If the authors have adequately addressed your comments raised in a previous round of review and you feel that this manuscript is now acceptable for publication, you may indicate that here to bypass the “Comments to the Author” section, enter your conflict of interest statement in the “Confidential to Editor” section, and submit your "Accept" recommendation.

Reviewer #1: All comments have been addressed

2. Is the manuscript technically sound, and do the data support the conclusions?

Reviewer #1: Yes

3. Has the statistical analysis been performed appropriately and rigorously? 

Reviewer #1: Yes

4. Have the authors made all data underlying the findings in their manuscript fully available?

Reviewer #1: Yes

5. Is the manuscript presented in an intelligible fashion and written in standard English?

Reviewer #1: Yes

6. Review Comments to the Author

Reviewer #1: The manuscript titled “Feasibility of the Xemio App for Breast Cancer Survivors in a Clinical Setting: Adherence, Acceptance, and Side Effect Monitoring (CTCAE vs. QoL)” is a well-constructed feasibility study evaluating a digital health tool for breast cancer survivors. The manuscript is clear, methodologically sound for a feasibility design, and addresses a relevant gap in mHealth survivorship tools. The integration of PROMs, CTCAE-based symptom reporting, and usability evaluation is well executed.

7. PLOS authors have the option to publish the peer review history of their article (what does this mean?). If published, this will include your full peer review and any attached files.

Reviewer #1: No

---

## [Editor Report · Acceptance letter]

PONE-D-25-09322R1

PLOS One

Dear Dr. Frid,

I'm pleased to inform you that your manuscript has been deemed suitable for publication in PLOS One. Congratulations! Your manuscript is now being handed over to our production team.

Kind regards,

on behalf of

Dr. Anton Pak

Academic Editor

PLOS One